# BIM: Block-Wise Local Learning with Masked Image Modeling

## Abstract

Like masked language modeling (MLM) in NLP, masked image modeling (MIM) extracts insights from image patches to enhance feature extraction in deep neural networks (DNNs). Unlike supervised learning, MIM pretraining requires substantial computational resources to handle large batch sizes (e.g., 4096), limiting its scalability. To address this, we propose Block-Wise Masked Image Modeling (BIM), which decomposes MIM tasks into sub-tasks with independent computations, enabling block-wise backpropagation instead of the traditional end-to-end approach. BIM achieves comparable performance to MIM while significantly reducing peak memory usage. For evaluation, we provide an anonymized GitHub repository here. Additionally, BIM facilitates concurrent training of multiple DNN backbones with varying depths, optimizing them for different hardware platforms while reducing computational costs compared to training each backbone separately.

## 1 Introduction

To enhance DNN performance, many current methodologies still rely heavily on supervised training paradigms, which demand a substantial amount of manual annotations. Recent advancements in self-supervised learning (SSL) offer an alternative solution with superior training accuracy and data efficiency. SSL leverages the inherent structures and patterns present in images, enabling the application of learned representations to specific target tasks using techniques like fine-tuning or linear classification. Among the diverse range of SSL strategies posited to date, Masked Image Modeling (MIM)-based approach He et al. (2022); Xie et al. (2022) has stood out due to its superior accuracy performance over the conventional supervised frameworks. MIM supervises the network by reconstructing occluded image patches by utilizing the visual representations from their visible counterparts. This paradigm intrinsically drives the encoders of Vision Transformer (ViT) to capture important features and patterns while discarding irrelevant or noisy information within an image.

However, MIM approaches have their downsides, as they necessitate substantial computational resources for processing large training data batches (e.g., 4096) with long training iterations (e.g., 800 epochs) He et al. (2022). This results in significantly higher memory usage and computational cost compared to conventional supervised learning, making the research and development in MIM prohibitively expensive in many scenarios. One way to attain efficient MIM training with reduced memory usage and computational requirements is to decompose the large DNN into smaller ones and train them separately in parallel. It has been shown in the previous literature that the biological brain is highly modular Caporale & Dan (2008), learning predominantly based on local information instead of a global objective that is optimized by backpropagating error signals Crick (1989); Marblestone et al. (2016). In light of those observations, an intuitive approach to alleviate memory usage of MIM entails dividing the model into multiple blocks, each trained independently. This approach, known as *block-wise local learning*, has the potential to substantially decrease memory requirements during training since memory space can be freed up as soon as training is completed for one block.

However, many existing local block-wise learning methods—specifically designed for another prominent self-supervised learning (SSL) approach, contrastive learning (CL) Lillicrap et al. (2014); Löwe et al. (2020); Xiong et al. (2020); Belilovsky et al. (2018)—have not been as successful in achieving performance comparable to end-to-end training. Unlike MIM-based approaches, contrastive learning methods Chen et al. (2020); Chen

Figure 1: Memory usage during forward and backward passes of DNN training.

& He (2020); Grill et al. (2020); He et al. (2020) leverage information from different views of each image to train the encoder, aiming to obtain high-quality representations for downstream tasks. Hence, the limitation of existing local block-wise learning methods is likely attributed to the fact that the learning objectives of supervised classification and contrastive learning rely heavily on global information. In contrast, MIM might only need local information to complete occluded patches at various masking ratios, which could lead to strong performance in blockwise learning on MIM.

Towards this end, we introduce *Block-Wise Masked Image Modeling* (BIM), which decomposes the global MIM task into several sub-tasks with independent compute patterns by implementing block-wise, rather than end-to-end, back-propagation operations. Within the BIM framework, each DNN block handles intermediate features derived from the preceding block, extracting features necessary for image reconstruction and block updates. The features that are extracted are subsequently employed to reconstruct the missing patches using a local decoder. Simultaneously, these extracted features are passed on to the next encoder block to continue their learning process.

Moreover, to further reduce the compute workload during the training process, we introduce an incremental masking strategy to spatially increase the masking ratio without compromising learning efficacy. Experiment results indicate that our proposed method can effectively reduce peak memory while maintaining model performance without extra computation overhead.

Aside from the memory efficiency advantages offered by BIM, it seamlessly incorporates the "Once-for-all" training paradigm Cai et al. (2019) through the simultaneous training of multiple ViT encoder backbones with increasing depths. This results in the generation of multiple trained DNN backbones of varying depth, each of which can adapt to different hardware platforms with unique computing capabilities. Consequently, this approach substantially diminishes computational expenses when compared to the individual training of each DNN backbone. Overall, our contributions are summarized as follows:

- We proposed BIM framework for memory and compute efficient SSL. BIM achieves a significant reduction in memory usage while delivering performance on par with traditional end-to-end training across a range of downstream tasks. Under the same batch size, BIM achieves approximately an average of 40% savings in peak memory and 80% in compute.

- BIM naturally enables multiple backbone DNNs with different depth to be trained jointly, yielding a set of pre-trained backbone DNNs that can be employed independently for subsequent tasks. Compared to training each backbone DNN separately, BIM offers added computational workload savings.

- To further reduce the computational cost, we introduce a novel MIM masking strategy that progressively increases the proportion of masked components over the input during BIM training, resulting in additional reductions in computational costs without losing in accuracy.

## 2 Background and Related Work

### 2.1 Memory Pattern during DNN Training

To provide a clearer insight into our approach, we will briefly outline the memory usage pattern during the training process of the DNNs. Figure 1 (a) illustrates the forward pass for a two-layer DNN. Throughout

the forward pass, intermediate activations are produced once the execution of each layer is completed. The intermediate activations need be stored in memory for gradient computation during the backward pass. The same process will repeat until the DNN output is produced. The storage of intermediate activations from all layers in memory is necessary for subsequent gradient calculations, leading to an increase in memory footprint until it reaches its peak value (Step 2 in Figure 1 (a)), and this peak memory usage increases in proportional with DNN depth.

Next, the DNN output generated during the forward pass is compared to the ground-truth value in the training dataset. Subsequently, the gradient is calculated and stored in memory for use in the forthcoming backward pass, which is depicted in Figure 1 (b). In the backward pass, the input activations of the last layer are initially retrieved from memory and used to compute the weight gradient by multiplication with the gradient. Additionally, the gradient is multiplied with the layer weights to generate the gradient for the preceding layers. Subsequently, the input activations and gradient can be removed, freeing up the corresponding memory space. This iterative process persists until gradients are computed for all the layer weights.

## 2.2 Self-supervised Learning

**Masked Image Modeling.** Inspired by the great success of MLM Ben Zaken et al. (2022); Bao et al. (2020); Sinha et al. (2021) in NLP, researchers have explored a similar masking methodology for computer vision tasks. The primary goal of MIM is to extract information from unmasked image patches in order to reconstruct the masked patches, performing MIM can significantly enhance the feature extraction capability of backbone DNN Xie et al. (2022). Existing research in the field has shown a preference for using large batch sizes (e.g., 4096), to train the backbone DNN with large number of epochs (e.g., 800), and has empirically observed improvements in MIM performance as a result. However, this places significant demands on memory capacity and compute capability. While some previous research has introduced advanced masking strategies to reduce the computation workload Kakogeorgiou et al. (2022); Li et al. (2022a); Liu et al. (2022); Kong & Zhang (2023); Wang et al. (2023), hardly any of these approaches have addressed the reduction of peak memory size. In this study, we employ the masked autoencoder (MAE) He et al. (2022), a pioneering framework for masked image modeling, as a benchmark to demonstrate how our method effectively reduces peak memory consumption and computational cost.

**Local Learning Paradigms.** Compared to end-to-end supervised learning approaches Dosovitskiy et al. (2021); He et al. (2016); LeCun & Bengio (1998), SSL methods Xie et al. (2022); He et al. (2022) require a considerably higher memory requirement due to larger batch size. Local learning has emerged as a practical strategy for addressing the issue of high memory demand. As a pioneering method for gradient-isolated block-wise training, the Greedy InfoMax algorithm Belilovsky et al. (2018) shows impressive performance on small datasets, using only 40% of the memory compared to what traditional supervised learning methods typically demand. However, the performance of InfoMax will degrade seriously on large-scale datasets, such as ImageNet. LoCo Xiong et al. (2020) is a local learning algorithm for unsupervised contrastive learning. By leveraging implicit gradient feedback between the gradient-isolation blocks, LoCo can achieve results comparable to those of the conventional contrastive learning framework. Nevertheless, LoCo comes with a significantly higher computational cost compared to the end-to-end training approach. In comparison, BIM enables a MIM-based SSL pretraining with substantial reductions in peak memory consumption and computational workload.

**Efficient DNN Training.** Previous research has explored various techniques to expedite DNN training through leveraging sparsity in weights and activations Mahmoud et al. (2020); Zhang et al. (2019); Yang et al. (2020); Choi et al. (2020); Qin et al. (2020); Zhang et al. (2022). For instance, Procrustes Yang et al. (2020) and Eager Pruning Zhang et al. (2019) improve training efficiency by aligning algorithms with hardware capabilities, eliminating unimportant DNN weights, and enhancing hardware efficiency. Another approach involves reducing DNN operand precision Judd et al. (2016); Lee et al. (2019) or dynamically adjusting precision during DNN training, as proposed in FAST Zhang et al. (2022). These techniques are orthogonal to our BIM framework and can provide additional training efficiency. There are also works focused on reducing memory consumption during DNN training by modifying the DNN architecture Gomez et al. (2017);

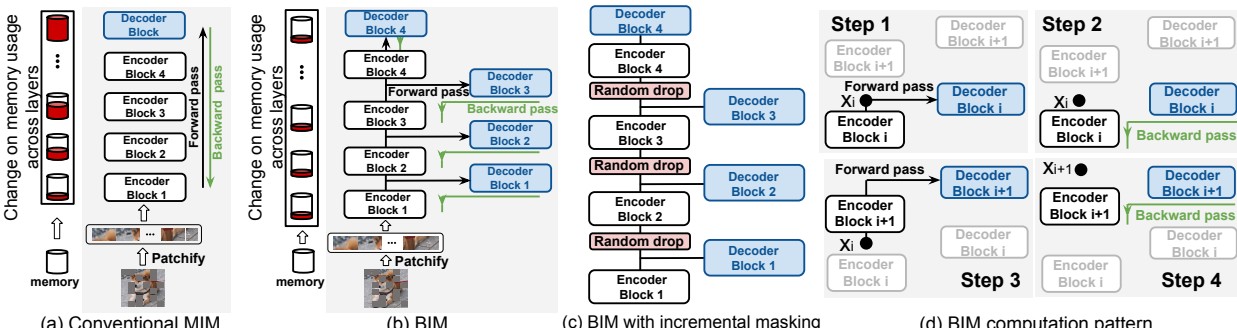

Figure 2: (a) Training flow for conventional MIM. (b) Training flow for BIM. (c) Overview of incremental masking strategy. (d) The computation pattern of BIM over two consecutive encoder blocks. The block dot denotes the intermediate activation during the forward pass.

Zhang et al. (2023). In comparison, BIM represents a versatile training framework applicable to all DNN architectures.

## 3 Method

In this section, we first introduce our memory-efficient BIM paradigm. We then provide a detailed explanation of the "Once-for-all" training approach for joint training across multiple backbone DNNs. Next, we discuss the incremental masking strategy to minimize computational overhead. Finally, we explain the memory scheduling technique that reduces memory consumption.

### 3.1 DNN Pre-training with BIM

The conventional MIM training process is illustrated in Figure 2 (a), where an end-to-end training algorithm is applied to all encoder and decoder blocks of ViT. This necessitates the allocation of extensive memory to store the entire set of model weights and gradients, resulting in a substantial demand for memory. To mitigate this, we proposed a block-wise local learning approach. An illustrative representation of our novel framework is presented in Figure 2 (b) and the pseudo-code of BIM is also provided in Algorithm 1. Before training, the stack of ViT encoders are initially divided into several blocks of uniform size, each associated with a decoder of the corresponding size. Subsequently, each ViT encoder-decoder pair undergoes separate training with the same objective loss function.

Specifically, the computational flow during the forward pass in BIM closely resembles that of MIM. In this process, each ViT encoder block receives intermediate results from the preceding blocks, conducts forward computations, and generates an output. This output is then duplicated: one copy is forwarded to the next encoder block for further processing, while the other copy is sent to the corresponding decoder block. The decoder's role is to produce predictions for masked patches using the features extracted by the current encoder block. This procedure continues until all the decoders have generated their predictions. The reconstructed patches from each decoder are subsequently compared to the original unmasked image patches, resulting in the generation of gradients for the backward pass operations.

With the gradients generated from the loss function at each decoder, the backward pass operations are carried out independently within each ViT encoder-decoder block. To be specific, the computation of gradients is terminated when it reaches the beginning of the current ViT encoder block, as shown by green arrows in Figure 2 (b). This ensures that there is no overlap in gradients across different blocks, and the gradient from one ViT encoder block does not influence the earlier ViT encoders. As a result of this gradient isolation strategy, in conjunction with our memory scheduling algorithm, BIM results in a significant reduction in peak memory usage than the conventional MIM.

### 3.2 Once-for-all Pre-training with BIM

In addition to the memory efficiency benefits provided by BIM, it also naturally implements the "Once-for-all" training paradigm Cai et al. (2019) by joint training of multiple ViT encoder backbones with growing depths. To illustrate this, if a ViT encoder is divided into four blocks, BIM allows the training of four distinct backbone DNNs with increasing depths by truncating at the output of each encoder block. This "Once-for-all" paradigm offered by BIM empowers the resultant pre-trained model to adapt to different computational constraints and diverse tasks, optimizing resource utilization and versatility. Compared to the conventional approach of separately training each backbone DNN with different depths, BIM results in substantial computational savings.

### 3.3 Incremental Masking Ratio Growth with BIM

We also introduce an incremental masking strategy that progressively increases the proportion of masked patches, aiming to achieve additional computational savings. Specifically, we enhance the level of difficulty in BIM by progressively decreasing the proportion of unmasked patches used for image reconstruction. As shown in Figure 2 (c), we introduce a new layer at the end of each encoder block, which randomly discards additional patches during the forward computation. The percentage of these additional drops is predefined and increases with the layer depth. This approach results in a reduction in the input size for each encoder block, effectively reducing the computational workload while obtaining a comparable (even better) performance as described in the experiment section.

### 3.4 BIM Computation Pattern

We describe an efficient computation pattern for both forward and backward passes of BIM training. The proposed computation pattern can achieve optimal peak memory consumption. We illustrate this scheduling algorithm with an example that show the computation pattern within the two consecutive encoder blocks (Figure 2 (d)).

Initially, the input patches $x_{i-1}$ are fed into the encoder block **i** to generate output $x_i$. All the intermediate activations, including $x_i$, are buffered in the memory for later use. (Step 1 in Figure 2 (d)). $x_i$ is subsequently input into decoder block **i**, resulting in the generation of predicted outputs and the initiation of the loss gradient. The backward pass starts as indicated in Step 2, results in additional weight updates in both decoder $i$ and encoder $i$. This backward pass concludes when it reaches the initial layer of encoder $i$. Once the parameter updates in encoder block **i** and decoder block **i** are finished, all intermediate features stored in the buffer, except for $x_i$, can be cleared from memory, preserving them for future use. $x_i$ is subsequently forwarded to encoder block **i+1**, and the identical process repeats, as illustrated in Steps 3 and 4 in Figure 2 (d).

## 4 Experiments

In this section, we conduct experiments to validate the BIM performance. We describe the implementation details and present the main results. The ablation studies are shown after the main results.

### 4.1 Implementation

**Pretrain model on ImageNet-1K** We evaluate BIM by comparing its performance against MAE He et al. (2022), and report its performance in terms of accuracy and peak memory consumption. We compare BIM with MAE over different versions of ViT, including ViT-base, ViT-large, and ViT-huge. The models are pretrained for either 400 or 800 epochs on the ImageNet-1k dataset Deng et al. (2009). We divide their encoder backbone into four blocks and train ViT-base, ViT-large and ViT-huge with a batch size of 4096 or 8192.

All the pretraining computations are executed on a GPU cluster consisting of 4 nodes, each node equipped with 8 NVIDIA V100 GPUs. We adopt a fixed masking ratio with a ratio of 75%, and we will test the

| Backbone | Patch Size | Pretrain Epoch | Pretrain Batch Size | Pretrain Method | Num of Blocks | Peak Memory Consumption (GB) | Fine-tuning Top-1 acc (%) | Linear Eval Top-1 acc (%) |
|---|---|---|---|---|---|---|---|---|
| ViT-base | $16^2$ | 400 | 4096 | MAE | - | 1218.57 (1×) | 83.01 | 63.03 |
| | | | | BIM | 4 | 929.09 (0.76×) | 82.98 ↓ 0.03 | 62.87↓ 0.16 |
| | | | 8192 | BIM | 4 | 1857.99 (1.52×) | 83.56↑ 0.55 | 65.46↑ 2.43 |
| | | 800 | 4096 | MAE | - | 1218.57 (1×) | 83.27 | 66.25 |
| | | | | BIM | 4 | 929.09 (0.76×) | 83.20↓ 0.07 | 66.08↓ 0.17 |
| | | | 8192 | BIM | 4 | 1857.99 (1.52×) | 83.89↑ 0.62 | 69.24↑ 2.99 |
| ViT-large | $16^2$ | 400 | 4096 | MAE | - | 1865.58 (1×) | 84.79 | 70.20 |
| | | | | BIM | 4 | 1093.51 (0.59×) | 84.58↓ 0.21 | 68.60↓ 1.60 |
| | | | 8192 | BIM | 4 | 2186.64 (1.17×) | 85.37↑ 0.58 | 74.07↑ 3.87 |
| | | 800 | 4096 | MAE | - | 1865.58 (1×) | 85.15 | 73.93 |
| | | | | BIM | 4 | 1093.51 (0.59×) | 84.99↓ 0.16 | 73.10↓ 0.83 |
| | | | 8192 | BIM | 4 | 2186.64 (1.17×) | 85.67↑ 0.52 | 76.51↑ 2.58 |
| ViT-huge | $14^2$ | 400 | 4096 | MAE | - | 3303.81 (1×) | 86.18 | 73.69 |
| | | | | BIM | 4 | 1802.40 (0.54×) | 86.10↓ 0.08 | 72.70↓ 0.99 |
| | | | 8192 | BIM | 4 | 3608.02 (1.09×) | 86.25↑ 0.27 | 74.81↑ 1.13 |
| | | 800 | 4096 | MAE | - | 3303.81 (1×) | 86.39 | 76.81 |
| | | | | BIM | 4 | 1802.40 (0.54×) | 86.24↓ 0.15 | 76.27↓ 0.54 |
| | | | 8192 | BIM | 4 | 3608.02 (1.09×) | 86.59↑ 0.20 | 77.60↑ 0.89 |

Table 1: Accuracy comparison for ViT backbones pretrained with BIM and MAE over Image classification task. BIM achieves a comparable accuracy performance as MAE while greatly saves the peak memory.

incremental masking ratio in the ablation study. We follow the original MAE work He et al. (2022) for the rest of the training settings. Pretrained ViT encoder backbones are evaluated over two downstream tasks, which are elaborated upon as follows.

**Image Classification.** We utilize pretrained ViT encoder backbones as the initializations. Subsequently, we append a linear layer to the pretrained encoder backbone and perform either end-to-end fine-tuning or linear probing on the ImageNet-1K dataset. In the case of end-to-end fine-tuning, the entire model undergoes fine-tuning, whereas for linear probing, only the weights of the linear layer are modified.

**COCO Detection and Instance Segmentation.** Additionally, we assess the performance of the pretrained backbone with tasks of object detection and instance segmentation. More precisely, we take Mask R-CNN He et al. (2017) as the objective detector and perform end-to-end finetuning on the COCO dataset Lin et al. (2015), among which the ViT backbone is adapted for use with FPN Lin et al. (2017), following ViTDet framework Li et al. (2022b). We apply this approach to all entries in Table 1.

## 4.2 Main Results

**Masked Image Reconstruction.** To evaluate the masked image reconstruction performance, we visualize some sample images that are taken from the validation set of ImageNet. The reconstruction visualization demonstrates that the ViT backbones trained with BIM can achieve a strong image reconstruction capability. The detailed reconstruction result comparison can be found in the appendix.

**End-to-End Fine-tuning.** For the end-to-end fine-tuning approach, we initiate the ViT by utilizing the pretrained encoder backbone and subsequently fine-tune the entire network. Specifically, we fine-tune ViT-base for 100 epochs and ViT-large and ViT-huge for 50 epochs, adhering to the procedures detailed in He et al. (2022). We assess the performance of our BIM with MAE in terms of accuracy and peak GPU memory consumption across multiple ViT backbones. The results are presented in Table 1, revealing some noteworthy findings. Firstly, under identical settings for training epochs and batch size, BIM consistently achieves comparable accuracies, with an average difference of less than 0.1%, in the end-to-end fine-tuning task across different backbone architectures. Secondly, when pretrained with a batch size of 8192, BIM attains an average fine-tuning accuracy that is 0.47 higher than the MAE that is pretrained with a batch size of 4096, across various DNN backbones.

| Pretrain Method | Pretrain Dataset | ViT-base | | ViT-large | |
|---|---|---|---|---|---|
| | | $AP^{box}$ | $AP^{mask}$ | $AP^{box}$ | $AP^{mask}$ |
| MAE | ImageNet | 51.2 | 45.5 | 54.6 | 48.6 |
| BIM | | 51.2 | 45.3 | 54.3 | 48.2 |

Table 2: Transfer learning results on **COCO** object detection and segmentation.

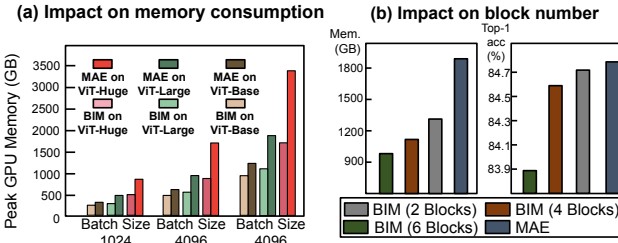

Figure 3: (a) Peak memory usage comparison with varying ViT backbones and batch sizes. (b) Memory and performance comparison with varying numbers of ViT blocks.

**Linear Probing.** In addition to the end-to-end fine-tuning, we also assess the linear probing scheme, where only the linear layer undergoes fine-tuning. As shown in Table 1, our proposed BIM demonstrates superior performance, exhibiting only an average accuracy drop of 0.52% when compared to the traditional MAE.

**Transfer Learning on COCO.** To assess the transferability of features derived from our proposed framework, we conduct end-to-end fine-tuning of Mask R-CNN for ViT on the COCO dataset using pretrained backbone weights from ViT-base and ViT-large, each pretrained for 800 epochs. Following the ViTDet implementation Li et al. (2022b), the ViT backbone is integrated with the FPN, applied uniformly across all pre-trained entities. The results, as shown in Table 2, demonstrate that ViT backbones pretrained with BIM can effectively transfer to object detection and instance segmentation tasks, yielding performance comparable to MAE.

**Transfer Learning on ADE20K.** To assess BIM's effectiveness in other common transfer learning tasks, we followed the experimental setup from the MAE paper and conducted experiments on ADE20KZhou et al. (2018) using UperNet. We fine-tuned the model end-to-end for 100 epochs with a batch size of 16, employing ViT-B and ViT-L pretrained on ImageNet-1K. Both ViT-B and ViT-L were divided into four blocks using BIM. MAE achieves mIoU scores of **47.7%** and **53.3%** on ViT-B and ViT-L, respectively, while BIM achieves **47.5%** and **53.0%**. The minimal performance difference on the ADE20K dataset indicates that BIM effectively preserves generalization ability without significant degradation.

**Saving on Peak Memory.** One of the key advantages offered by BIM is the substantial reduction in peak GPU memory usage. To visualize this, Figure 3 (a) provide a peak memory consumption comparison between MAE and BIM across various batch size, under the settings that all the ViT backbones are divided into four blocks for BIM training. We notice that BIM achieves an average of 25%, 42% and 48% peak memory savings than MAE on ViT-base, ViT-large and ViT-huge, respectively. Specifically, the peak memory savings increase as the ViT encoder backbone size increases. This is due to the diminishing impact of the ViT decoder on memory usage as the encoder size grows. Ignoring the impact of other components such as decoders and embedding layers, BIM training of a ViT backbone with four blocks using BIM can result in up to a $4\times$ reduction in peak memory usage. BIM offers a promising solution for researchers facing constraints in GPU resources. Specifically, BIM enables a ViT backbone to be trained with a larger batch size, and therefore leading to better accuracies over the downstream tasks.

**Performance under the Same Peak Memory.** In this section, we evaluate the performance of BIM and MAE while maintaining a fixed peak memory consumption. To achieve this, we adjust the batch size during BIM training on ViT-large so that its peak memory usage matches that of MAE with a batch size of 4096. This results in a batch size of 6784 for both ViT-large. We then train ViT-large with BIM for 400

| Pretrain Method | Backbone | Acc with Num of Blocks (%) | | | | Training Cost Saving (%) |
|---|---|---|---|---|---|---|
| | | 1 | 2 | 3 | 4 | |
| IT | | 70.59 | 79.04 | 81.48 | 83.01 | - |
| MD | ViT-base | 65.88 | 75.02 | 79.46 | 83.01 | 61.12 |
| BIM | | 70.59 | 79.06 | 81.39 | 82.98 | **68.00** |
| IT | | 77.78 | 82.49 | 84.22 | 84.79 | - |
| MD | ViT-large | 73.70 | 79.41 | 82.78 | 84.79 | 78.23 |
| BIM | | 77.78 | 82.3 | 84.14 | 84.58 | **85.18** |
| IT | | 80.75 | 84.88 | 85.53 | 86.12 | - |
| MD | ViT-huge | 75.78 | 80.34 | 83.48 | 86.12 | 80.85 |
| BIM | | 80.75 | 84.52 | 85.50 | 86.03 | **87.84** |

Table 3: Fine-tuning performance comparison among IT, MD, and BIM. BIM performs much better than MD, while achieving a comparable performance as IT. Compared with IT, BIM significantly reduces training cost by up to 87.84%.

| Method | Mask Ratio (%) | | | | | Fine-tune Top-1 Acc (%) |
|---|---|---|---|---|---|---|
| | Block 1 | Block 2 | Block 3 | Block 4 | On Average | |
| Fixed Ratio | 60 | | | | | 81.79 |
| | 70 | | | | | 83.80 |
| | 75 | | | | | 84.58 |
| | 80 | | | | | 84.28 |
| | 90 | | | | | 82.67 |
| Incremental Masking Ratio | 75 | 80 | 85 | 90 | 82.5 | 84.44 |
| | 65 | 70 | 80 | 85 | 75 | **84.75** |

Table 4: Comparison between fixed masking ratio and incremental masking ratio.

epochs using a batch size of 6784 and finetune it over the ImageNet dataset with 50 epochs. This leads to an accuracy of 85.02%, which is better than that obtained with the MAE backbone (84.79%).

**Once-for-all Training of BIM.** As discussed in the method section, BIM offers a natural advantage by allowing the simultaneous training of multiple ViT-encoder backbones with varying depths, all sharing their weights in a nested manner. Our evaluation of ViT encoders' performance is conducted on the ImageNet dataset. Specifically, we compare our approach with two alternative baseline methods. The first baseline, referred to as *independent training (IT)*, involves training each ViT encoder backbone with varying depth separately. Note that this approach incurs significantly higher training costs since each ViT encoder is trained independently. Here we compute the overall parameters to be trained for generating four models as the training cost. The second baseline, denoted as *MAE directly (MD)*, entails truncating the pretrained MAE at the end of each ViT encoder block and fine-tuning it for downstream tasks. All ViT backbones are pretrained for 400 epochs, then fine-tuned with 100 epochs for ViT-base and 50 epochs for ViT-large and ViT-huge.

As indicated in Table 3, we note that BIM outperforms MD by a significant margin and closely matches the performance of IT. In particular, for ViT with a single encoder block, BIM and IT exhibit identical training schemes, resulting in the same accuracy. Notably, BIM achieves an average of 77% reduction in training cost compared to IT.

### 4.3 Ablation Study

**Impact of Masking Ratio.** We evaluate the performance of BIM over different masking ratios. Specifically, we want to investigate two problems. Firstly, we examine how BIM accuracy is influenced by different masking ratios, assuming all the blocks have the identical masking ratio. Secondly, we investigate how accuracy varies when adjusting the masking ratio across different ViT encoder blocks.

We trained with the ViT-large encoder backbone using various masking ratios for 400 epochs and subsequently fine-tuned on the ImageNet dataset for 50 epochs. As shown in Table 4, we observe that under the assumption that all the blocks have the same masking ratio, a masking ratio of 75% yields the best overall performance, which aligns with findings from the MAE approach He et al. (2022). Furthermore, it is worth noting that incremental masking ratios in general outperforms constant masking ratios under the same average masking ratio. In particular, we find that masking ratios of 65%, 70%, 80%, 85%, with an average masking ratio of 75%, yield superior performance compared to a fixed masking ratio of 75% across all blocks. In comparison, employing a combination of masking ratios such as 75%, 80%, 85%, and 90% with an average masking ratio

| Method | Batch Size | Peak Memory | Fine-tuning Accuracy |
|---|---|---|---|
| SimMIM | 4096 | 880.91GB | 83.07% |
| BIM+SimMIM | 4096 | 231.46GB | 83.02% |
| BIM+SimMIM | 8192 | 462.56GB | 83.58% |

Table 5: Performance of BIM with SimMIM Pretraining on ViT-L with 400 epochs.

| Method | Batch Size | Peak Memory | Fine-tuning Accuracy |
|---|---|---|---|
| BIM | 4096 | 1093.51GB | 84.58% |
| MAE (8-bit) | 4096 | 590.06GB | 62.59% |
| MAE (16-bit) | 4096 | 945.42GB | 78.73% |
| MAE (20-bit) | 4096 | 1165.87GB | 82.29% |

Table 6: BIM and quantization training on ViT-L.

of 82.5% results in an accuracy of 84.51%. This performance is comparable to that achieved with an average masking ratio of 75%, while greatly reducing computational workload.

**Impact of Block Numbers.** In this section, we explore how the number of ViT blocks affects the accuracy of BIM. We investigate how varying the number of blocks impacts BIM's performance. To achieve this, we pretrained the ViT-large backbone with 400 epochs using 2 blocks, 4 blocks, and 6 blocks, followed by fine-tuning the entire ViT-large model on ImageNet with 50 epochs.

The findings presented in Figure 3 (b) reveal that a larger number of blocks indeed results in lower peak memory usage for BIM. However, this reduction in memory consumption comes at the cost of decreased model performance. For example, when transitioning from 4 to 6 blocks, BIM achieves only a marginal 4% reduction in memory consumption while incurring a notable 0.7% drop in accuracy, which may not be justified.

**Impact on Decoder Architecture.** In BIM, we follow the original decoder design specified in the MAE work by setting the decoder depth to 8. Additionally, BIM is compatible with other decoder architectures. For instance, BIM can also integrate with SimMIM Xie et al. (2022), which employs a simple multilayer perceptron (MLP) as its decoder. We evaluate the performance of BIM over the SimMIM framework and the results are depicted in Table 5. BIM, when combined with SimMIM, not only significantly lowers peak memory usage but does so without a notable compromise in accuracy. Moreover, it is worth highlighting that BIM leads to an improvement in SimMIM's performance, a 0.51% increase in accuracy, by doubling the batch size from 4096 to 8192, while still achieving a much smaller peak memory than the original SimMIM.

**Impact on Training time** As a trade-off, BIM inevitably incurs additional computation costs due to the added decoder blocks after each encoder branch. For instance, pretraining ViT-L with MAE for 400 epochs takes approximately 35 hours using 32 V100 GPUs. In contrast, BIM pretraining, due to the added decoder blocks in each encoder block, takes 39 hours. However, with the incremental masking technique, BIM's pretraining time reduces to 37 hours. This reduction occurs because incremental masking decreases the input size for each encoder block, effectively reducing the computational workload. Additionally, for other types of masked pretraining schemes with a light decoder architecture, such as SimMIM, the increase in training time is minimal, the results are shown in Table 7 for different batch size. For example, pretraining ViT-L with SimMIM for 400 epochs takes approximately 27 hours with BIM and 26 hours without BIM, both using 32 V100 GPUs. Finally, we would like to highlight that BIM is designed to mitigate the memory constraints present in MAE-based pretraining, making MAE feasible on memory-limited devices.

**Comparison between BIM and Quantized Pretraining.** Before BIM, efficient DNN training methods like parameter quantization were used to reduce memory consumption but had notable drawbacks. For

| Pretrain Method | ViT-base | ViT-large | ViT-huge |
|-----------------|----------|-----------|----------|
| SimMIM | 17h | 26h | 59h |
| BIM + SimMIM | 17h | 27h | 60h |

Table 7: Training time of BIM on SimMIM with 400 epochs using a batch size of 4096.

| Batch Size | 1024 | 2048 | 4096 |
|------------|------|------|------|
| ViT-base | **82.11**/82.18 | **82.50**/82.57 | **82.98**/83.01 |
| ViT-large | **83.85**/83.90 | **84.16**/84.30 | **84.58**/84.79 |
| ViT-huge | **85.32**/85.34 | **85.71**/85.79 | **86.10**/86.18 |

Table 8: Performance comparison of BIM (bold) and MAE (regular) across different batch sizes.

example, quantization often led to significant performance drops. Table 6 compares the fine-tuning accuracies of BIM and quantized MAE pretraining. We quantized all gradients, activations, and weights using a linear scheme during MAE pretraining and fine-tuned the model on ImageNet. Pretraining MAE with 20-bit quantization resulted in higher peak memory (1165.87 GB) but lower accuracy (82.29%), while 8-bit quantization reduced memory but severely degraded accuracy (62.59%) compared to BIM (84.58%). As discussed in the methods section, BIM can be integrated with these techniques to further reduce memory usage while maintaining superior accuracy.

**Impact on large batch size** We evaluate the performance of BIM across different batch sizes. The accuracy results for models of varying sizes and batch sizes are presented in the table below, where bold numbers indicate BIM's results, and regular-weight numbers represent MAE's results. As shown in the table 8, BIM achieves performance comparable to MAE across diverse batch sizes.

**Impact on different MIM method** To evaluate BIM on other MIM methods, we applied it to MaskFeat with HOG features Dalal & Triggs (2005), using ViT-Base and ViT-Large to assess BIM's effectiveness. The results show that MaskFeat achieves an accuracy of **83.80%** on ViT-B and **85.83%** on ViT-L. With BIM applied, the accuracy remains high at **83.72%** on ViT-B and **85.64%** on ViT-L, further demonstrating BIM's strong potential when integrated with MaskFeat.

**Impact on Training Iteration.** Finally, we investigate the effect of training iterations on accuracy during the fine-tuning phase for downstream tasks. We train ViT-large using BIM and MAE for varying numbers of epochs: 400, 800, 1200, and 1600, then fine-tune on the ImageNet for 50 epochs. The accuracy results, as shown in Table 9, indicate a general trend of increasing accuracy with training iteration for both BIM and MAE. Nevertheless, it is worth noting that the incremental gain in accuracy is relatively modest when extending training from 800 epochs to 1600 epochs.

## 5 Conclusion

MIM pretraining typically demands significant computational resources and memory requirements, especially when handling large training data batches. In this work, we introduce BIM as an alternative solution, achieving comparable performance to MIM while significantly reducing peak memory consumption. This opens up interesting future avenues in a promising direction of research. Future works can be conducted on integrating BIM with other self-supervised learning frameworks such as iBOT Zhou et al. (2021) or PECO Dong et al. (2023), etc.

## References

Masked autoencoder official implementation. *https://github.com/facebookresearch/mae/*, 2021.

| Iterations | 400 epochs | 800 epochs | 1200 epochs | 1600 epochs |
|---|---|---|---|---|
| BIM Acc. (%) | 84.58 | 85.09 | 85.15 | 85.27 |
| MAE Acc. (%) | 84.79 | 85.15 | 85.23 | 85.38 |

Table 9: Experiment on increasing pretraining epochs. Both MAE and BIM benefit from longer pretraining epochs.

Hangbo Bao, Li Dong, Furu Wei, Wenhui Wang, Nan Yang, Xiaodong Liu, Yu Wang, Jianfeng Gao, Songhao Piao, Ming Zhou, and Hsiao-Wuen Hon. UniLMv2: Pseudo-masked language models for unified language model pre-training. In Hal Daumé III and Aarti Singh (eds.), *Proceedings of the 37th International Conference on Machine Learning*, volume 119 of *Proceedings of Machine Learning Research*, pp. 642–652. PMLR, 13–18 Jul 2020. URL `https://proceedings.mlr.press/v119/bao20a.html`.

Eugene Belilovsky, Michael Eickenberg, and Edouard Oyallon. Greedy layerwise learning can scale to imagenet. In *International Conference on Machine Learning*, 2018. URL `https://api.semanticscholar.org/CorpusID:57189514`.

Elad Ben Zaken, Yoav Goldberg, and Shauli Ravfogel. BitFit: Simple parameter-efficient fine-tuning for transformer-based masked language-models. In *Proceedings of the 60th Annual Meeting of the Association for Computational Linguistics (Volume 2: Short Papers)*, pp. 1–9, Dublin, Ireland, May 2022. Association for Computational Linguistics. doi: 10.18653/v1/2022.acl-short.1. URL `https://aclanthology.org/2022.acl-short.1`.

Han Cai, Chuang Gan, Tianzhe Wang, Zhekai Zhang, and Song Han. Once-for-all: Train one network and specialize it for efficient deployment. *arXiv preprint arXiv:1908.09791*, 2019.

Natalia Caporale and Yang Dan. Spike timing–dependent plasticity: a hebbian learning rule. *Annu. Rev. Neurosci.*, 31:25–46, 2008.

Ting Chen, Simon Kornblith, Mohammad Norouzi, and Geoffrey Hinton. A simple framework for contrastive learning of visual representations, 2020. URL `https://arxiv.org/abs/2002.05709`.

Xinlei Chen and Kaiming He. Exploring simple siamese representation learning, 2020. URL `https://arxiv.org/abs/2011.10566`.

Seungkyu Choi, Jaehyeong Sim, Myeonggu Kang, Yeongjae Choi, Hyeonuk Kim, and Lee-Sup Kim. An energy-efficient deep convolutional neural network training accelerator for in situ personalization on smart devices. *IEEE Journal of Solid-State Circuits*, 55(10):2691–2702, 2020.

Kevin Clark, Minh-Thang Luong, Quoc V. Le, and Christopher D. Manning. Electra: Pre-training text encoders as discriminators rather than generators, 2020.

Francis Crick. The recent excitement about neural networks. *Nature*, 337(6203):129–132, 1989.

N. Dalal and B. Triggs. Histograms of oriented gradients for human detection. In *2005 IEEE Computer Society Conference on Computer Vision and Pattern Recognition (CVPR'05)*, volume 1, pp. 886–893 vol. 1, 2005. doi: 10.1109/CVPR.2005.177.

Jia Deng, Wei Dong, Richard Socher, Li-Jia Li, Kai Li, and Li Fei-Fei. Imagenet: A large-scale hierarchical image database. In *2009 IEEE Conference on Computer Vision and Pattern Recognition*, pp. 248–255, 2009. doi: 10.1109/CVPR.2009.5206848.

Xiaoyi Dong, Jianmin Bao, Ting Zhang, Dongdong Chen, Weiming Zhang, Lu Yuan, Dong Chen, Fang Wen, Nenghai Yu, and Baining Guo. Peco: Perceptual codebook for bert pre-training of vision transformers. In *Proceedings of the AAAI Conference on Artificial Intelligence*, volume 37, pp. 552–560, 2023.

Alexey Dosovitskiy, Lucas Beyer, Alexander Kolesnikov, Dirk Weissenborn, Xiaohua Zhai, Thomas Unterthiner, Mostafa Dehghani, Matthias Minderer, Georg Heigold, Sylvain Gelly, Jakob Uszkoreit, and Neil Houlsby. An image is worth 16x16 words: Transformers for image recognition at scale. *ICLR*, 2021.

Golnaz Ghiasi, Yin Cui, Aravind Srinivas, Rui Qian, Tsung-Yi Lin, Ekin D. Cubuk, Quoc V. Le, and Barret Zoph. Simple copy-paste is a strong data augmentation method for instance segmentation, 2021.

Aidan N Gomez, Mengye Ren, Raquel Urtasun, and Roger B Grosse. The reversible residual network: Backpropagation without storing activations. *Advances in neural information processing systems*, 30, 2017.

Priya Goyal, Piotr Dollár, Ross Girshick, Pieter Noordhuis, Lukasz Wesolowski, Aapo Kyrola, Andrew Tulloch, Yangqing Jia, and Kaiming He. Accurate, large minibatch sgd: Training imagenet in 1 hour, 2018.

Jean-Bastien Grill, Florian Strub, Florent Altché, Corentin Tallec, Pierre H. Richemond, Elena Buchatskaya, Carl Doersch, Bernardo Avila Pires, Zhaohan Daniel Guo, Mohammad Gheshlaghi Azar, Bilal Piot, Koray Kavukcuoglu, Rémi Munos, and Michal Valko. Bootstrap your own latent: A new approach to self-supervised learning, 2020. URL https://arxiv.org/abs/2006.07733.

Kaiming He, Xiangyu Zhang, Shaoqing Ren, and Jian Sun. Deep residual learning for image recognition. In *2016 IEEE Conference on Computer Vision and Pattern Recognition (CVPR)*, pp. 770–778, 2016. doi: 10.1109/CVPR.2016.90.

Kaiming He, Georgia Gkioxari, Piotr Dollár, and Ross Girshick. Mask r-cnn. In *Proceedings of the IEEE international conference on computer vision*, pp. 2961–2969, 2017.

Kaiming He, Haoqi Fan, Yuxin Wu, Saining Xie, and Ross Girshick. Momentum contrast for unsupervised visual representation learning, 2020. URL https://arxiv.org/abs/1911.05722.

Kaiming He, Xinlei Chen, Saining Xie, Yanghao Li, Piotr Dollár, and Ross Girshick. Masked autoencoders are scalable vision learners. In *2022 IEEE/CVF Conference on Computer Vision and Pattern Recognition (CVPR)*, pp. 15979–15988, 2022. doi: 10.1109/CVPR52688.2022.01553.

Patrick Judd, Jorge Albericio, Tayler Hetherington, Tor M Aamodt, and Andreas Moshovos. Stripes: Bit-serial deep neural network computing. In *Microarchitecture (MICRO), 2016 49th Annual IEEE/ACM International Symposium on*, pp. 1–12. IEEE, 2016.

Ioannis Kakogeorgiou, Spyros Gidaris, Bill Psomas, Yannis Avrithis, Andrei Bursuc, Konstantinos Karantzalos, and Nikos Komodakis. What to hide from your students: Attention-guided masked image modeling. In *Computer Vision – ECCV 2022*, pp. 300–318. Springer Nature Switzerland, 2022. ISBN 978-3-031-20056-4. doi: 10.1007/978-3-031-20056-4_18. URL https://link.springer.com/chapter/10.1007/978-3-031-20056-4_18.

Xiangwen Kong and Xiangyu Zhang. Understanding masked image modeling via learning occlusion invariant feature. In *Proceedings of the IEEE/CVF Conference on Computer Vision and Pattern Recognition (CVPR)*, pp. 6241–6251, June 2023.

Yann LeCun and Yoshua Bengio. *Convolutional Networks for Images, Speech, and Time Series*, pp. 255–258. MIT Press, Cambridge, MA, USA, 1998. ISBN 0262511029.

Jinsu Lee, Juhyoung Lee, Donghyeon Han, Jinmook Lee, Gwangtae Park, and Hoi-Jun Yoo. 7.7 lnpu: A 25.3 tflops/w sparse deep-neural-network learning processor with fine-grained mixed precision of fp8-fp16. In *2019 IEEE International Solid-State Circuits Conference-(ISSCC)*, pp. 142–144. IEEE, 2019.

Gang Li, Heliang Zheng, Daqing Liu, Chaoyue Wang, Bing Su, and Changwen Zheng. Semmae: Semantic-guided masking for learning masked autoencoders. *arXiv preprint arXiv:2206.10207*, 2022a.

Yanghao Li, Hanzi Mao, Ross Girshick, and Kaiming He. Exploring plain vision transformer backbones for object detection, 2022b.

Timothy P. Lillicrap, Daniel Cownden, Douglas B. Tweed, and Colin J. Akerman. Random feedback weights support learning in deep neural networks, 2014.

Tsung-Yi Lin, Michael Maire, Serge Belongie, Lubomir Bourdev, Ross Girshick, James Hays, Pietro Perona, Deva Ramanan, C. Lawrence Zitnick, and Piotr Dollár. Microsoft coco: Common objects in context, 2015.

Tsung-Yi Lin, Piotr Dollár, Ross Girshick, Kaiming He, Bharath Hariharan, and Serge Belongie. Feature pyramid networks for object detection, 2017.

Zhengqi Liu, Jie Gui, and Hao Luo. Good helper is around you: Attention-driven masked image modeling, 2022.

Ilya Loshchilov and Frank Hutter. Sgdr: Stochastic gradient descent with warm restarts, 2017.

Ilya Loshchilov and Frank Hutter. Decoupled weight decay regularization, 2019.

Sindy Löwe, Peter O'Connor, and Bastiaan S. Veeling. Putting an end to end-to-end: Gradient-isolated learning of representations, 2020.

Mostafa Mahmoud, Isak Edo, Ali Hadi Zadeh, Omar Mohamed Awad, Gennady Pekhimenko, Jorge Albericio, and Andreas Moshovos. Tensordash: Exploiting sparsity to accelerate deep neural network training. In *2020 53rd Annual IEEE/ACM International Symposium on Microarchitecture (MICRO)*, pp. 781–795. IEEE, 2020.

Adam H Marblestone, Greg Wayne, and Konrad P Kording. Toward an integration of deep learning and neuroscience. *Frontiers in computational neuroscience*, 10:94, 2016.

Eric Qin, Ananda Samajdar, Hyoukjun Kwon, Vineet Nadella, Sudarshan Srinivasan, Dipankar Das, Bharat Kaul, and Tushar Krishna. Sigma: A sparse and irregular gemm accelerator with flexible interconnects for dnn training. In *2020 IEEE International Symposium on High Performance Computer Architecture (HPCA)*, pp. 58–70. IEEE, 2020.

Koustuv Sinha, Robin Jia, Dieuwke Hupkes, Joelle Pineau, Adina Williams, and Douwe Kiela. Masked language modeling and the distributional hypothesis: Order word matters pre-training for little. In *Proceedings of the 2021 Conference on Empirical Methods in Natural Language Processing*, pp. 2888–2913, Online and Punta Cana, Dominican Republic, November 2021. Association for Computational Linguistics. doi: 10.18653/v1/2021.emnlp-main.230. URL https://aclanthology.org/2021.emnlp-main.230.

Christian Szegedy, Vincent Vanhoucke, Sergey Ioffe, Jonathon Shlens, and Zbigniew Wojna. Rethinking the inception architecture for computer vision, 2015.

Haochen Wang, Kaiyou Song, Junsong Fan, Yuxi Wang, Jin Xie, and Zhaoxiang Zhang. Hard patches mining for masked image modeling. In *Proceedings of the IEEE/CVF Conference on Computer Vision and Pattern Recognition (CVPR)*, pp. 10375–10385, June 2023.

Zhenda Xie, Zheng Zhang, Yue Cao, Yutong Lin, Jianmin Bao, Zhuliang Yao, Qi Dai, and Han Hu. Simmim: A simple framework for masked image modeling. In *International Conference on Computer Vision and Pattern Recognition (CVPR)*, 2022.

Yuwen Xiong, Mengye Ren, and Raquel Urtasun. Loco: Local contrastive representation learning. In *Proceedings of the 34th International Conference on Neural Information Processing Systems*, NIPS'20, Red Hook, NY, USA, 2020. Curran Associates Inc. ISBN 9781713829546.

Dingqing Yang, Amin Ghasemazar, Xiaowei Ren, Maximilian Golub, Guy Lemieux, and Mieszko Lis. Procrustes: a dataflow and accelerator for sparse deep neural network training. In *2020 53rd Annual IEEE/ACM International Symposium on Microarchitecture (MICRO)*, pp. 711–724. IEEE, 2020.

Yang You, Igor Gitman, and Boris Ginsburg. Large batch training of convolutional networks, 2017.

Sangdoo Yun, Dongyoon Han, Seong Joon Oh, Sanghyuk Chun, Junsuk Choe, and Youngjoon Yoo. Cutmix: Regularization strategy to train strong classifiers with localizable features, 2019.

Hongyi Zhang, Moustapha Cisse, Yann N. Dauphin, and David Lopez-Paz. mixup: Beyond empirical risk minimization, 2018.

Jiaqi Zhang, Xiangru Chen, Mingcong Song, and Tao Li. Eager pruning: algorithm and architecture support for fast training of deep neural networks. In *2019 ACM/IEEE 46th Annual International Symposium on Computer Architecture (ISCA)*, pp. 292–303. IEEE, 2019.

Sai Qian Zhang, Bradley McDanel, and HT Kung. Fast: Dnn training under variable precision block floating point with stochastic rounding. In *2022 IEEE International Symposium on High-Performance Computer Architecture (HPCA)*, pp. 846–860. IEEE, 2022.

Sai Qian Zhang, Thierry Tambe, Nestor Cuevas, Gu-Yeon Wei, and David Brooks. Camel: Co-designing ai models and embedded drams for efficient on-device learning. *arXiv preprint arXiv:2305.03148*, 2023.

Bolei Zhou, Hang Zhao, Xavier Puig, Tete Xiao, Sanja Fidler, Adela Barriuso, and Antonio Torralba. Semantic understanding of scenes through the ade20k dataset, 2018. URL https://arxiv.org/abs/1608.05442.

Jinghao Zhou, Chen Wei, Huiyu Wang, Wei Shen, Cihang Xie, Alan Yuille, and Tao Kong. ibot: Image bert pre-training with online tokenizer. *arXiv preprint arXiv:2111.07832*, 2021.

# Appendix

# A  Implementation Details

In this section, we will provide an in-depth overview of the specific configurations utilized for both the pretraining and finetuning phases of BIM.

## A.1  BIM pseudo code for one iteration

The pesudo code of BIM for one iteration is provided below.

---
**Algorithm 1** BIM pseudo code for one iteration

---
```
# N: Number of encoder-decoder blocks      f_n: n-th encoder block
# h_n: n-th decoder block      L_n: Loss of block n      M: Loss function
for im, mask in Dataloader do
    x_0 = patchify(im) # patchify image into patches
    for n in range(N) do
        x_{n+1} = f_n(x_n, mask) # send the masked input x to f_n and obtain x_{n+1}.
        im_{n+1} = h_n(x_{n+1}, mask) # reconstruct im
        L_n = M(im, im_{n+1}, mask)
        L_n.backward() # backprop across f_n and h_n
        update(f_n, h_n) # update weights in f_n, h_n
        Keep x_{n+1} and free the memory of gradient and activations in f_n and h_n.
    end for
    Release all the memory.
end for
```

---

## A.2  BIM architecture

Following the original MAE, we implemented the standard ViT architecture as the backbone architecture. In MAE, it sets the encoder and decoder to have different widths and adopts a linear projection layer to match them. Besides, its encoder also ends with a LayerNorm. Hence, to match this setting, for all blocks in BIM, we assigned the same LayerNorm and linear layer to match them with their identical decoder. Except for this difference, all other architecture settings are the same with raw MAE.

### A.3 Pretraining Settings

Before pretraining, following official ViT implementation, we used xavier_uniform in Pytorch to initialize all Transformer blocks. Besides, we used a conventional linear learning rate ($lr$) scaling rule: $lr = base\_lr \times batch\_size/256$. Notably, the batch size is selected from 2048, 4096, and 8192, according to the experiment requirement. Each ViT backbone is pretrained with either 400 epochs or 800 epochs. Other general pretraining configurations are listed in Table 10.

| General Configuration | Detail |
|---|---|
| Optimizer | AdamW Loshchilov & Hutter (2019) |
| Base Learning Rate | 1.5e-4 |
| Weight Decay | 0.05 |
| Optimizer Momentum | $\beta_1, \beta_2 = 0.9, 0.95$ |
| Learning Rate Schedule | cosine decay Loshchilov & Hutter (2017) |
| Warmup Epochs Goyal et al. (2018) | 40 |
| Data Augmentation | RandomResizedCrop |

Table 10: General configuration for the pretraining process.

### A.4 End-to-end Fine-tuning Settings

The pretrained ViT encoder backbone with BIM are fully-finetuned over ImageNet dataset. The end-to-end fine-tuning training epochs is set to 100 for ViT-base, and 50 for ViT-large and ViT-huge. Besides, the value of the drop path is set to be 0.1, 0.2 and 0.3 for ViT-base, ViT-large and ViT-huge, respectively. Base learning rate is set to 5e-4 for ViT-base and 1e-3 for ViT-large and ViT-huge. The other general configurations we used are listed in Table 11.

| General Configuration | Detail |
|---|---|
| Optimizer | AdamW |
| Weight Decay | 0.05 |
| Optimizer Momentum | $\beta_1, \beta_2 = 0.9, 0.999$ |
| Layer-wise $lr$ Decay Clark et al. (2020) | 0.75 |
| Batch Size | 1024 |
| Learning Rate Schedule | cosine decay |
| Augmentation | RandAug (9, 0.5) |
| Label Smoothing Szegedy et al. (2015) | 0.1 |
| Mixup Zhang et al. (2018) | 0.8 |
| Cutmix Yun et al. (2019) | 1.0 |

Table 11: General configuration for the end-to-end fine-tuning process.

### A.5 Linear Probing Settings

Following the linear probing implementation in the original MAE, we disabled many common regularization strategies and set the weight decay to 0. The other general configurations are listed in the Table 12.

| General Configuration | Detail |
|---|---|
| Optimizer | LARS You et al. (2017) |
| Base learning rate | 0.1 |
| Optimizer momentum | 0.9 |
| Batch size | 16384 |
| Learning rate schedule | Cosine decay |
| Warmup epochs | 10 |
| Training epochs | 90 |
| Augmentation | RandomResizedCrop |

Table 12: General configuration for the linear probing process.

### A.6 Settings for Once-for-all BIM Training

For *Independent training (IT)*, we employ a procedure where we selectively truncate the ViT encoder backbone to different depths. Subsequently, we conduct MAE pretraining for each of these truncated subnetworks. Following the pretraining phase, we proceed with end-to-end finetuning for each of these pretrained ViT backbones. All ViT backbones are pretrained for 400 epochs, then fine-tuned with 100 epochs for ViT-base and 50 epochs for ViT-large and ViT-huge. The rest settings align with those elaborated in setting sections. Following this, we proceed to perform end-to-end fine-tuning for each truncated ViT backbone. Specifically, we employ 100 epochs for ViT-base and 50 epochs for both ViT-large and ViT-huge during the fine-tuning process.

### A.7 Object Detection and Segmentation in COCO

As we introduced in 4.1, following ViTDet framework Li et al. (2022b), we adapt the vanilla ViT for the use of an FPN Lin et al. (2017) backbone in MaskR-CNN He et al. (2017). The size of the input image is $1024 \times 1024$, augmented with large-scale jittering Ghiasi et al. (2021) during training. We fine-tune both the ViT-base and ViT-large backbone 100 epochs with the AdamW optimizer. Notably, the feature map scale for both backbone architectures is set to 1/16, , stride=16, since the patch size is 16. We report box AP for object detection and mask AP for instance segmentation. The value of the drop path is 0.1 for ViT-base and 0.4 for ViT-large. Other general configurations are listed in Table 13.

### A.8 Peak Memory Measurement

We used a Python library **torch-summary** to obtain the peak GPU memory consumption of MAE and BIM with batch sizes of 2048, 4096, and 8192, separately. Torch-summary is a Python library that provides a

| General Configuration | Detail |
|---|---|
| Optimizer | AdamW |
| Optimizer Momentum | $\beta_1, \beta_2 = 0.9, 0.999$ |
| Warmup Iterations | 250 |
| Batch size | 64 |
| Base learning rate | 0.1 |
| Augmentation | Large-scale Jitter |
| Scale Range | [0.1,2.0] |
| Input Size | $1024 \times 1024$ |
| Learning Rate | 1e-4 |
| Weight Decay | 0.1 |
| Epochs | 100 |

Table 13: General configuration for the transfer learning in COCO.

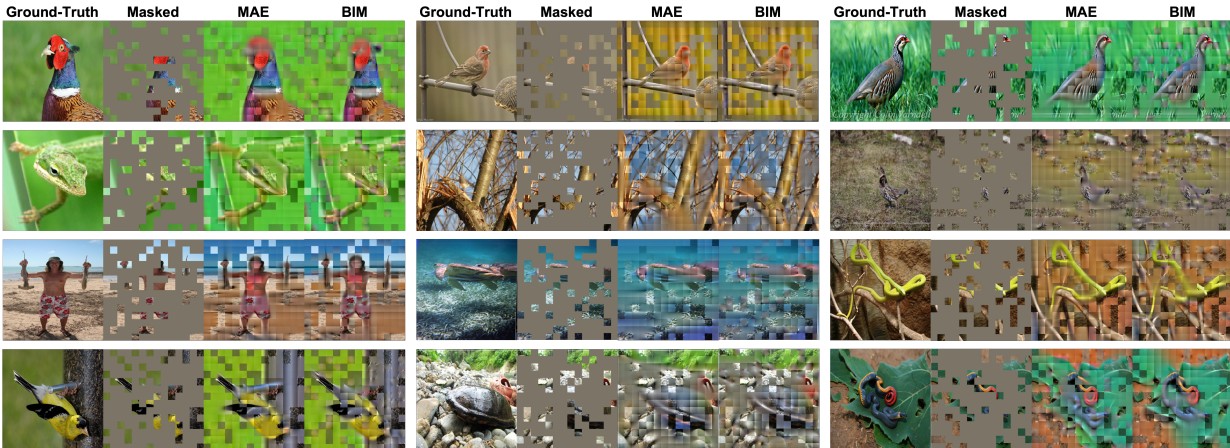

Figure 4: Results on image reconstruction. For each set of four images, we display them in the following order from left to right: ground-truth, masked image, MAE reconstruction, and our BIM reconstruction. The backbone DNN is extracted from ViT-large model pretrained with either BIM or MAE with 800 epochs. The weights for the MAE backbone is downloaded from the official website mae (2021).

convenient way to quickly collect the information about a PyTorch neural network model. It allows you to quickly see the peak memory usage during the training process of the model.

## B Masked Image Reconstruction

As Figure 4 shows, for each quadruple, from left to right, we present the ground-truth images, the images with 75% masking, the MAE reconstruction results, and BIM reconstruction results, respectively. The reconstruction visualization demonstrates that the ViT backbones trained with BIM can achieve a strong image reconstruction capability.

