# OpenReview forum: "BIM: Block-Wise Local Learning with Masked Image Modeling"
_TMLR — Rejected by TMLR_

### Review · Reviewer_ZfMA · 2025-07-28

**Summary Of Contributions:**

This paper proposes Block-Wise Masked Image Modeling (BIM) to improve memory efficiency in Masked Image Modeling (MIM) pretraining. The main contributions are:
1. A BIM framework that decomposes MIM tasks into sub-tasks with independent compute patterns, enabling block-wise backpropagation instead of end-to-end training
2. Achieving comparable performance to MAE while reducing peak memory usage by an average of 40%
3. Natural support for "Once-for-all" training paradigm, enabling simultaneous training of multiple backbones with varying depths
4. Introduction of an incremental masking strategy to further reduce computational costs

Strengths:
- Practical approach enabling MIM pretraining in memory-constrained environments
- Thorough experimental validation across multiple datasets (ImageNet, COCO, ADE20K)
- Less performance degradation compared to existing methods like quantization

Weaknesses:
- Unclear premise that MIM requires large batch sizes (unlike contrastive learning, MIM has no cross-batch interactions)
- Increased computation and training time due to additional decoders at each block
- Contradiction between claimed 80% compute savings and actual training time increases

**Audience:**

Yes

**Audience Explanation:**

This technique could be useful for researchers who need to train large models in memory-constrained environments. Specifically:
1. Enables MIM pretraining in GPU-limited research settings
2. Once-for-all training efficiently produces multiple models for different hardware platforms
3. Minimal overhead when combined with lightweight decoders like SimMIM

However, considering current GPU memory improvements and the performance-time tradeoffs, practical applicability may be limited.

**Broader Impact Concerns:**

No specific ethical concerns. This research focuses on technical efficiency improvements with no anticipated negative societal impacts.

**Claims And Evidence:**

No

**Claims Explanation:**

The paper's main claims are partially contradictory or insufficiently supported:
1. Lack of evidence for large batch size requirements: The authors claim MIM requires large batch sizes (e.g., 4096), but theoretically, MIM doesn't require large batches as it lacks cross-batch interactions unlike contrastive learning. Table 8 results also show performance is maintained with smaller batch sizes.
2. Contradictory computational cost claims: The introduction claims 80% compute savings, but experiments show training time increases from 35 to 39 hours due to additional decoders per block. Even with incremental masking, it's still 37 hours.
3. Insufficient analysis of performance-memory-time tradeoffs: While memory decreases, training time increases and performance slightly drops. This tradeoff lacks thorough analysis.
4. Missing theoretical explanation for incremental masking: No intuitive explanation for why deeper layers can tolerate more masking.

**Requested Changes:**

Critical Changes:
1. Provide clear justification for MIM's large batch size requirements: Add experimental evidence or theoretical explanation for why MIM needs large batch sizes, or revise this claim.
2. Accurate analysis of computational costs: Remove or clarify the "80% compute savings" claim. Present consistent FLOPs calculations and training time measurements.
3. Add performance-memory-time tradeoff analysis: Clearly analyze the tradeoffs between memory savings, performance degradation, and training time increases. Specify situations where BIM would be beneficial.

Strengthening Changes:
1. Add theoretical justification for incremental masking strategy: Provide intuitive explanation for why deeper layers can handle more masking.
2. Experiments with varying decoder depths: Current decoder depth is fixed at 8. Explore shallower decoders to potentially reduce computational overhead.
3. Comparison with other efficient training methods: Add comparisons with other memory-saving techniques like gradient checkpointing and mixed precision training.
4. Expand conclusion section: The current conclusion is too brief. Need deeper discussion of contributions, limitations, and future research directions.

---

### Review · Reviewer_Jasv · 2025-08-05

**Summary Of Contributions:**

The paper proposed Block-Wise Masked Image Modeling (BIM) for masked image modeling (MIM), where it decomposes MIM tasks into sub-tasks with independent computations, thus reducing the peak memory usage.

**Audience:**

Yes

**Audience Explanation:**

The paper belongs to low-resource training for masked image modeling, and it will be attractive to a large group of audience.

**Broader Impact Concerns:**

No broader impact concerns.

**Claims And Evidence:**

Yes

**Claims Explanation:**

The experimental results demonstrated clear memory saving when using BIM, and the gap between BIM and MAE on fine-tuning top-1 accuracy and linear probe top-1 accuracy is small, given the same batch size. If the batch size of BIM is increased to match the memory usage of MAE, the performance could be a little bit better. The performance is also validated on downstream tasks like COCO detection and segmentation tasks.

**Requested Changes:**

1. The selection and comparison of the incremental masking ratio is not extensively studied. In Table 4, the authors only provide two sets of incremental mask settings. I would like to see a more comprehensive study of the incremental masking ratio, given different increment steps and different average masking ratios. In addition, are there any principal ways to select the incremental masking ratio?

2. The proposed method BIM seems a naive combination of local training + MAE/MIM, and the method section is short and contains few methodological designs. I hope the authors could explicitly emphasize what is the difference between BIM and previous local training methods, as shown in their related works

3. The current BIM uses complete local training at the beginning. If the number of blocks is too large ($\leq$ 6), then the performance degradation becomes obvious. Is it possible to gradually reduce the number of blocks during training to further improve the performance? In addition, since the training is decomposed into sub-tasks, is it possible to further reduce the training time to random drop backward calculation for different blocks?

Minor:

4. Please resize the tables in the appendix to improve readability and ensure consistent formatting.

Points 1 and 2 are critical to securing my recommendation for acceptance. Point 3 is good to have and could be a plus to the work.

---

### Review · Reviewer_whYs · 2025-08-12

**Summary Of Contributions:**

This paper proposes a new paradigm for masked image modeling by divide models into sub-blocks. The pretraining process is processed with forward and backward stages within each block. All these designs help reduce peak memory usage. Authors have conducted extensive experiments to show the competitive performance, scalability and resources efficiency. The proposed paradigm has potentials to advance current majority of DNN backbone pretraining recipes.

**Audience:**

No

**Audience Explanation:**

I have some concerns of this section, some of them are also explained in the "Requested Changes" section as well.

i am afraid that this work might not interest enough audience, given advances in [1] (cited by the authors), [2], [3], which have explored both autoencoder, ViT and CLIP pretraining.

Besides, results of Table 1 are not convinced, see discussions in the "Requested Changes".

In addition, scalability with data is not clear.

[1]. Masked Autoencoders Are Scalable Vision Learners
[2]. Scaling Language-Image Pre-training via Masking
[3]. RECLIP: Resource-efficient CLIP by Training with Small Images

**Broader Impact Concerns:**

I donot have broader impact concerns.

**Claims And Evidence:**

Yes

**Claims Explanation:**

Overall, I think the logic of the paper is clear and the motivation of this work is well explained. This paper is driven by how to better reduce peak memory usage as well as maintain good performance and scalability. Method section is explained clearly on how to train models with masked image modeling. Experiments are conducted extensively. Authors select MAE as the baseline and compare experimental results on various model architecture, showing peak memory usage gains and performance quality.

**Requested Changes:**

I have some concerns to be addressed further.

1. The module "Incremental masking ratio" is less convincing to me, it is very likely another implementation of masking strategy which has been widely used in large-scale pretrainings.

2. I think thought the peak memory usage has been reduced. The overall training flow is more complicated and not as elegant as the E2E-manner. Dividing blocks, have intermediate copies and multiple backward pass are less feasible, less simple and less implementation-friendly.

3. Results in Table 1, since the performance is not changing linearly with Batch size, the comparison is not convinced to me. I would suggest authors compare MAE with BIM under the same peak memory usage.

4. Besides, I wonder how BIM will help with data scaling, I am concerned that the capabilities of BIM will be limited with significant increasing of data amount.

5. For Table 2, I would recommend authors add memory usage as well. Otherwise, I donot see significance by showing this comparisons without any advantages.

---

### Decision · Action_Editor_bdZd · 2025-10-10

**Recommendation:** Reject

**Audience:**

Yes

**Audience Explanation:**

The topic is relevant. However, the work would only be of interest to TMLR's audience if the findings and claims are well justified.

**Claims And Evidence:**

No

**Claims Explanation:**

A number of concerns were raised but they were not addressed by the authors. Some major concerns include:
* Lack of evidence for large batch size requirements
* Contradictory computational cost claims
* Insufficient analysis of performance-memory-time tradeoffs
* Missing theoretical explanation for incremental masking
* Unclear on selection of incremental masking ratio
* Novelty of BIM when compared with previous local training methods
* Feasibility of the current local training scheme when the number of blocks is large

**Resubmission Of Major Revision:**

The authors may consider submitting a major revision at a later time.